# *Mindmaster Roleplay*:
# A SOCIAL REASONING AND PLANNING BENCHMARK

## ABSTRACT

Social intelligence is one of the most challenging capabilities to develop in Artificial Intelligence (AI) systems. Existing benchmarks for social reasoning mainly rely on unstructured text dialogues or simplified scenarios. There are very limited platforms that can support the community to systematically investigate the complex social cognitive mechanisms in social interactions. Thus, we present *Mindmaster Roleplay*, a social interaction platform that captures the dynamic interplay between beliefs, intentions, values, and actions through dyadic role-play games. Our platform provides interpretable first-person annotations of mental states, enabling researchers to trace how reasoning evolves and influences decision-making in diverse social scenarios. Our dataset establishes a valuable foundation for training and evaluating AI systems that more closely resemble human social intelligence in complex social reasoning tasks. Our experiments and analyses with both LLMs and human participants reveal a range of intriguing phenomena in social reasoning and decision-making. We will release our platform, dataset, code, and models upon acceptance.

## 1 INTRODUCTION

Human intelligence is primarily distinguished by our exceptional social cognitive abilities, which far exceed those of other primates (Herrmann et al., 2007). As AI systems advance, developing socially intelligent agents has emerged as a critical frontier (Cassell, 2022; Fan et al., 2022). A genuinely socially intelligent agent must perform online, context-sensitive social reasoning (Tomasello, 2010; 2019) and adaptive planning(Tambe, 1997; Conte et al., 1995), which means to infer diverse mental states under complex, dynamic environments while regulating its own intentions and actions based on intrinsic values and resolving internal conflicts. These constitute core capabilities for human-like social interactions characterized by agility and flexibility (Dautenhahn, 1995; Apperly, 2010), but continue to pose substantial challenges for modern AI systems.

One major challenge in building socially intelligent AI systems lies in the dual nature of the environment: Beyond the observable physical world, there is also an invisible but significant mental world, just like "dark matter" (Zhu et al., 2020). For example, latent mental states (e.g., beliefs, intentions, and values, etc.), different from visible signals like gaze directions, gestures, or actions, play key roles in shaping decisions and actions (Tomasello, 2010). A socially intelligent agent must perceive and predict many variables within a vast state space, reason over the graph formed by visible and latent variables, and subsequently make decisions (Tomasello, 2019; Fan et al., 2022). A central challenge in AI is to develop a unified computational framework in which structured mental representations can emerge, evolve, and guide behavior in interpretable, human-aligned ways through efficient reasoning and planning across both physical and mental domains.

*Cognitive architecture* provides a unified computational framework for modeling human-like cognitive mechanisms (Laird et al., 2017; Langley et al., 2009). However, traditional cognitive architectures (Anderson & Lebiere, 1998) primarily focus on individual cognitive processes such as memory and learning, with very limited attention paid to social cognition (Kotseruba & Tsotsos, 2020). Others have begun to integrate multiple cognitive components to support social interaction, but usually focus on a single dominant component (Vinanzi & Cangelosi, 2024) or depend on handcrafted rules that limit scalability and adaptability (Clodic et al., 2018; Sun, 2016). The well-known Belief-Desire-Intention (BDI) framework (Bratman, 1987; Georgeff et al., 1999) incorporates diverse cog-

nitive elements, but is typically centered on individual decision-making (Baker et al., 2017) and lacks support for recursive and nested mental state reasoning. Tomasello (2010) proposed a theoretical dual-agent communication model highlighting intent communication, but it lacks integration with other cognitive components. Recent rapid development of Large Language Models (LLM) also gave birth to many LLM social agents (Chen et al., 2025; Xu et al., 2024; Hu & Ying, 2025; Zhou et al., 2023). Yet, these models often rely on dialogues, surface heuristics, and struggle with generalization or recursive reasoning (Wang et al., 2025; Vinanzi & Cangelosi, 2024). A unified cognitive architecture that systematically integrates multiple cognitive components for rich and dynamic social interaction remains an open challenge.

Existing benchmarks further limit progress, relying on synthetic textual QA tasks (mostly focusing on isolated mental states and Sally-Anne-style (Baron-Cohen et al., 1985) simple tasks) that neglect multi-turn dynamics and rich structured mental-state trajectories (Chen et al., 2024; Lee et al., 2025), and fail to capture the complexity of real-world social cognition. Many rely on LLM-generated dialogues verified ex post by humans (Zhou et al., 2023; Yu et al., 2025), creating a distributional gap between first-person enactment and third-person verification. Few resources support structured, interactive, extensible environments where beliefs, intents, values, and actions can fully interact with each other together and evolve with the social context (Hu & Ying, 2025; Vinanzi & Cangelosi, 2024).

To address these limitations, we introduce *Mindmaster Roleplay*, the first general-purpose, physically simulated, and cognitively grounded platform for studying multi-agent social interaction. Grounded in the BDI framework (Bratman, 1987; Georgeff et al., 1999) and Tomasello's theory of agent communication (Tomasello, 2010), our platform specifically targets the core mental mechanisms underlying dual-agent interaction. Unlike existing approaches that isolate specific cognitive processes, *Mindmaster Roleplay* supports two core cognitive processes: inverse reasoning (inferring others' minds) and forward planning (determining one's own minds and actions), as well as three core cognitive components: belief, intent, and value. Going beyond simple false-belief tasks like the Sally-Anne test (Baron-Cohen et al., 1985), we design a rich suite of cognitively grounded scenarios—drawing from classic psychological paradigms (Tomasello, 2010; Melis & Tomasello, 2019), which capture complex social dynamics such as cooperation, helping, competition, deception, concealment, harm, and strategic counter-deception, etc. Moreover, *Mindmaster Roleplay* is carefully designed with a generalizable object space featuring everyday items selected to represent diverse affordances, along with a structured action, intent, and value space. This setup enables both humans and models to engage in long-term social interactions, while naturally eliciting first-person annotations, reasoning, and explanations of mental states and actions throughout gameplay, thereby mitigating the gap between first-person "ground-truth" and third-person labels of mental states, and reducing annotation noise, establishing better evaluation criteria for social intelligence. Another advantage is that we don't simply study a single mental component; we study human cognitive mechanisms from the perspective of the entire cognitive architecture. Therefore, we can systematically and comprehensively study the dependencies and associations between mental variables, such as how values influence intent and how beliefs influence intent. Our interactive game environment is extensible and supports human–human, human-AI, and AI–AI configurations, making it a promising platform for future "social Turing test"–style evaluations.

Building on *Mindmaster Roleplay*, we conducted a large-scale human-subject study with 232 participants in controlled, in-person experimental sessions. To ensure high-quality data, we implemented a comprehensive onboarding process with a tutorial and mandatory quiz. Only participants who completed this process participated in the interaction phase, where they engaged in gameplay with assigned initial intent and value profiles. During interaction, participants performed social reasoning and planning while annotating their mental states and estimating their partner's mental states. Participants also provided natural language explanations of their reasoning processes, creating interpretable cognitive traces valuable for model training and evaluation. After extensive data cleaning and preprocessing, we produced a high-quality dataset of annotated human social interactions. While smaller than datasets generated by Large Language Models (LLMs) (Zhou et al., 2023) or rule-based simulations (Sclar et al., 2024), our dataset offers superior cognitive fidelity and human alignment. It captures the full spectrum of mental dynamics, including belief updates, partner mind modeling, intention updates, and value-driven trade-offs throughout naturalistic dyadic interactions.

Beyond data collection, we performed comprehensive experiments and analyses to uncover insights into human cognitive mechanisms during social interaction. We evaluated state-of-the-art large lan-

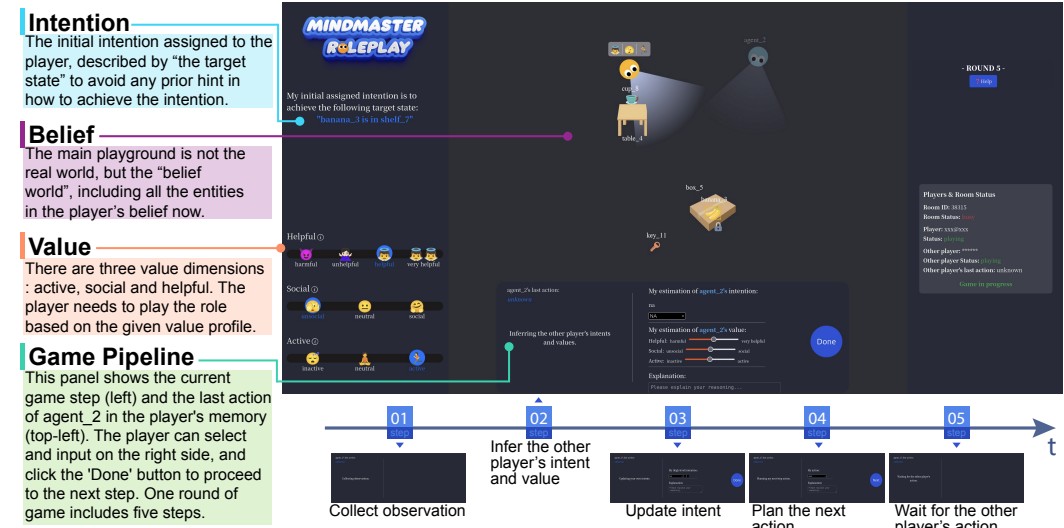

Figure 1: **Game Interface and Game Flow of Mindmaster Roleplay Platform.** (1) **Observation**: The sector depicted represents the agent's attention direction and field of view, encompassing the entire area between the two boundary rays. (2) **Belief**: Highlighted objects indicate those currently observed by the agent, while grayed objects represent the agent's memory of previously perceived positions stored in belief. (3) **Values**: The platform incorporates three value dimensions: (i) active dimension (inactive/neutral/active), (ii) social dimension (unsocial/neutral/social), and (iii) helpful dimension (harmful/unhelpful/neutral/helpful/very helpful). Each value state corresponds to a distinct icon. (4) **Intention**: Participants are assigned an initial intent. (5) **Game Pipeline**: Each round consists of five sequential stages derived from cognitive processes based on the BVI framework and Tomasello's theory: (i) collecting observations, (ii) inferring other player's intent and values, (iii) updating intent, (iv) planning the next action, and (v) awaiting the other player's action. Note that there are **no given scripts** during the game, and agents are required to perform freely based on the assigned initial intent and value profile.

guage models and humans, fine-tuned a large language model, and built a cognitive agent architecture. We find the "uncertain" nature of human cognition, the strengths and weaknesses of current large language models, and the potential of building agent architectures.

In summary, our contributions include:

- A cognitively grounded, dyadic interaction platform *Mindmaster Roleplay* for studying human-like social reasoning and planning;
- A high-quality dataset from 232 human participants, annotated with fine-grained mental states and reasoning traces from first-person views;
- Comprehensive experiments and analyses of current popular methods and humans, demonstrating our dataset's value for training and evaluation.

## 2 PLATFORM AND DATA COLLECTION

### 2.1 THE *Mindmaster Roleplay* PLATFORM

**Cognitive Process Decomposition** The essence of social cognition is learning a decision function $P(a_t|o_{0:t})$ that maps observations ($o_{0:t}$) to actions ($a_t$). Following the Bayesian Theory of Mind (BToM) framework, we decompose this process into modular components representing distinct cognitive functions. We propose that an agent's action planning $P(a_t)$ depends directly on three mental states: intention $\eta_t$ (target state the agent aims to achieve), belief $b_t$ (the agent's understanding of the world and others), and value $v$ (the agent's stable preferences). Note that "value" is conceptually similar to "desire" in the BDI (Bratman, 1987; Georgeff et al., 1999) framework; we adopt the term "value" instead of "desire" to better align with the terminology commonly used in the current AI community. Formally, this gives us $P(a_t|\eta_t, b_t, v)$. The belief state $b_t$ is nested and encompasses belief about the world states and the others' mental states $b_t = (b(s_t), b(b'_t), b(\eta'_t), b(v'))$, where the prime notation denotes the other agent's mental states. The intention is updated based on belief, value, and previous intent: $P(\eta_t|b_t, v, \eta_{t-1})$. We assume that the agent's value $v$ remains stable

Figure 2: **Components of the Game. Here we only present a subset of examples** for each component; please refer to the appendix for the complete space. Several scenarios are derived from established experiments in cognitive psychology, enabling systematic investigation of specific cognitive functions. Additional scenarios are procedurally generated according to predefined environmental parameters and task constraints. Here we show five typical scenario examples: (1) cooperate to open the box using a key; (2) help to find something; (3) understand compositional pointing gestures; (4) play chess together; (5) pointing gesture disambiguation.

throughout the game interaction. Given these components, we can systematically factorize the decision function:

$$P(a_t|v, \eta_{t-1}, b_{t-1}, o_t) \tag{1}$$

$$= \sum_{\eta_t, b_t} P(a_t|\eta_t, b_t, v) P(\eta_t|\eta_{t-1}, b_t, v) P(b(b'_t), b(\eta'_t), b(v')|b(s_t), b_{t-1}) P(b(s_t)|b_{t-1}, o_t) \tag{2}$$

This modular approach allows us to model each cognitive process independently while preserving their functional integration in the complete social cognitive architecture. Please refer to the supplementary material for the complete derivation of the equations.

**The Platform Design**    Based on the cognitive process decomposition described above, our game environment is structured into several key sub-steps: collecting observations (i.e., $o_t$), inferring the other player's intent and value (i.e., $P(b(b'_t), b(\eta'_t), b(v')|b(s_t), b_{t-1})$), updating the agent's own intent (i.e., $P(\eta_t|\eta_{t-1}, b_t, v)$), planning the next action (i.e., $P(a_t|\eta_t, b_t, v)$), and waiting for the other player's action. The agent is required to complete these steps sequentially during the game by selecting corresponding options from dropdown menus, while also articulating the reasoning and justification for each choice in natural language within a designated text box. See Figure 1 for an illustration.

As shown in Figure 1, the central playground in our game interface represents the agent's belief world, rather than the real world. The platform adopts a "dark-room spotlight + memory residue" metaphor to imitate the belief world, in which the agent has partial observations of the world, and only entities in the agent's belief are shown in the belief world. Among these entities in belief, the entities currently within the agent's perception field appear in vivid colors, while entities that were previously observed and stored in memory are rendered in faded tones, indicating residual memory traces. We have curated diverse interactive objects with distinct affordances (*e.g.*, banana (eatable), cabinet (container), chess (joint activity), etc.) and implemented atomic actions (*e.g.*, wave hand, drink, eat, point to, nod head, grab, etc.) based on involved affordances and daily life interactions. A scenario is defined by the initial spatial configuration of objects and agents, as well as their pre-assigned intentions and values. Our scenario design enables complex interactions and positions previous experimental paradigms (*e.g.*, (Melis & Tomasello, 2019)), allowing us to both reproduce existing findings and generate novel interactions. Based on these objects and actions, we systematically define an intent space encompassing all possible agent intentions. Figure 2 shows some intent examples, such as "find", "get", "put onto", "open", "inform", "help", "request help", "harm", etc. As for value, we have three value dimensions significant for social interaction: "active" (preference for physical motion), "social" (preference for social interaction) and "helpful" (preference for assisting others). Each value dimension in our framework has multiple possible levels, as illustrated in

Figure 3: **Data Collection Process**: Step 1 participant recruitment; Step 2 participants training; Step 3 participants playing game in pairs; Step 4 preprocessing raw data to get high-quality clean data

Figure 1 with corresponding icons. Note that the choice of value dimension is not arbitrary: These three value dimensions cover the typical values involved in basic single-agent activities ("active") as well as higher-level dyadic social activities ("social" and "helpful"). Moreover, these dimensions may come into conflict, providing a useful setting to study the mechanisms by which agents make decisions under value conflicts. During gameplay, each agent is shown its own value profile, and the associated icons are persistently displayed around the agent. This visual cue serves as a constant reminder for the agent to stay immersed in its assigned role and make decisions aligned with the designated value profile. Figure 2 shows an illustration of object, action, intent spaces, and scenario examples. Please see our supplementary materials for more details about our game designs (*e.g.*, the scenario designs, details of all spaces, our game website link, etc.)

After completing the tutorial and passing the quiz, each participant is paired with another player to enter the game. Players are instructed to role-play based on their assigned initial intent (if not "None") and value configuration, and sequentially complete the cognitive sub-steps shown in Figure 1. The gameplay unfolds as turn-based interactions over a fixed number of rounds. A session ends when a player successfully fulfills their objective or the maximum round limit is reached. The experimental design may introduce tensions between value-driven dispositions and intention fulfillment, encouraging participants to engage in context-sensitive reasoning and planning from their character's perspective. These decision points reveal how agents prioritize competing motivations and reconcile potential conflicts, such as: 1) Whether to abandon personal goals to assist others; 2) How to proceed when experiencing low motivation but facing goal requirements; and 3) When to seek assistance versus pursuing independent action. Through this methodology, we examine how agents with different mental state configurations infer others' mental states, update their own mental models, resolve value conflicts, and plan appropriate actions in social contexts. Note that we also ask the players to provide their reasons for all steps so as to assist future model training and evaluation.

## 2.2 DATASET COLLECTION AND ANALYSES

The detailed illustration of the data collection process can be seen at Fig. 3. The main game is conducted in pairs, with each participant paired with another. Participants would be assigned an initial intent (including intent none) and value, and if necessary, they are encouraged to reasonably adjust their intent based on their character's role. They are asked to embody their assigned roles, reason thoughtfully according to the given context, and make coherent decisions accordingly.

Based on the process described, *Mindmaster Roleplay* provides a unique opportunity to collect unobservable data on human social interactions. While some may argue that explicitly asking individuals to mark their intentions may introduce biases compared to spontaneous thought processes, this approach offers a valuable alternative in the absence of direct brain imaging techniques like fMRI. It

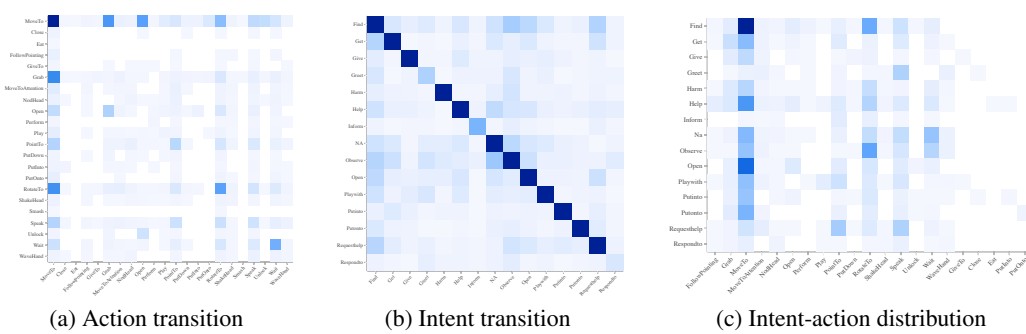

| (a) Action transition | (b) Intent transition | (c) Intent-action distribution |

Figure 4: Heatmaps for action transition, intent transition and intent-action distribution.

stands as one of the most effective ways to collect data on human cognitive processes, facilitating significant advances in the field of social cognition and human intention understanding.

### 2.3 DATA STATISTICS AND EXAMPLES

Basic statistics indicate our dataset maintains balance in initial intent and value distribution, as detailed in the supplementary material. Figure 4 illustrates significant patterns in action and intent transitions. The action transition matrix reveals that "MoveTo" and "RotateTo" constitute the majority of action pairs, serving as physical prerequisites for subsequent operations, while other frequently transitioning actions include object interactions ("Grab", "Open") and agent-to-agent interactions ("PointTo", "Speak"). Intent transition analysis demonstrates that most intents persist across rounds, with exceptions being the quickly resolvable temporal intents "Greet", "Inform", and "RespondTo"; notably, participants executing intents such as "Find", "Get", and "Open" exhibit higher probability of requesting assistance. The intent-action distribution matrix reveals structural relationships between these dimensions: "MoveTo" correlates strongly with most intents as a fundamental prerequisite action, the high co-occurrence between "PointTo" and "RequestHelp" demonstrates how physical gestures effectively convey assistance needs, and certain actions appear exclusively with specific intents (*e.g.*, action "PutInto" with intent "PutInto"), indicating partial overlap between intention and action spaces.

Generally speaking, the collected human data is diverse and reflects human decisions, especially when there are conflicts in assigned Value and Initial Intent. As we limit the verbal communications, the non-verbal way of communicating is accentuated: how to convey your intent non-verbally and how the other comprehends the information. The agent is supposed to make various choices according to the attributed values, though in the same scenario.

In Figure 5, we provide two examples of the human social interactions in our dataset. The key frames from the two agents' perspectives are shown. The agents' action strategies are consistent with their assigned value. Meanwhile, the other agent's intent and value are successfully inferred from observation. More examples can be seen in the appendix.

## 3 EXPERIMENTS AND ANALYSES

### 3.1 EXPERIMENT SETUP

**Task Decomposition** Based on the factorization in Equation equation 2, we can decompose the social cognitive process into four distinct subtasks:

- **Intention and Value Estimation** $P(b(v', \eta'_t)|b(s_t))$: This task involves inferring others' underlying intentions and values from their observed behaviors. We introduce two observational conditions: partial observation (where the agent has limited information as would occur in actual interactions) and full observation (where complete world information is available). This distinction enables us to determine whether inference errors stem from information limitations or from constraints in the agent's inferential capabilities.

- **Intention Updating** $P(\eta_t|\eta_{t-1}, b_t, v)$: Revising one's own intentions in response to current beliefs and value systems, reflecting adaptations environmental and others' mental changes.

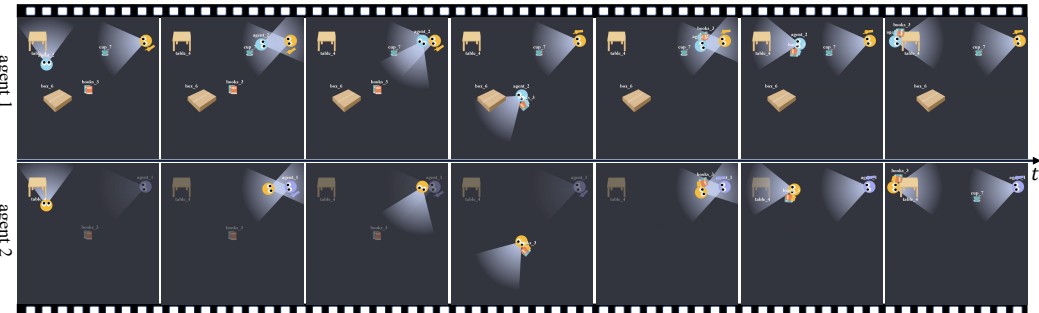

(a) Agent 1 initially engages in cooperative play chess to pursue his own task, but upon inferring Agent 2's intent for getting a cup (from the performing drinking action), he leaves and smashes the cup. Agent 2 observes this behavior and correctly infers Agent 1's harmful value as well as his intent of harm.

(b) Agent 1 (inactive value) points to the book, which Agent 2 (active, helpful value) interprets as a request and retrieves it. Subsequently, from Agent 1's pointing to the table, Agent 2 correctly infers the intent to place the book on the table and successfully assists Agent 1 in completing the task.

Figure 5: Qualitative examples of the human social interactions in our dataset. We only select key frames from the whole videos.

- **Social Interaction Policy** $P(a_t|\eta_t, b_t, v)$: Generating appropriate responses by integrating updated intentions, current beliefs, and contextual observations.

This decomposition enables systematic evaluation of models' social intelligence by isolating key cognitive components, thereby facilitating precise benchmarking and targeted improvements in computational social cognition capabilities.

**Evaluation Metrics**  Representing answer with $q$ and the human label with $p$, we employ the following metrics to assess the performance of LLMs across the four tasks:

- **Similarity:** For selection tasks, we measure whether any of the model's top-$n$ predicted outcomes match the human labels: $Sim_n = \sum_{i=1}^{N} \sum_{j=1}^{n} I(q_{i,j} == p_i)/N$. This approach provides a more comprehensive evaluation beyond single-prediction accuracy.

- **Confidence Discrimination (Cfd):** We calculate the difference between the average confidence scores for correct and incorrect predictions: $Cfd = \sum_{\{q|q==p\}} cf(q)/N(q == p) - \sum_{\{q|q!=p\}} cf(q)/N(q! = p)$. The confidence is reported by the model itself. A larger Cfd value indicates superior ability to discriminate between correct and incorrect responses with self report confidence, reflecting better calibration.

- **Value Distance:** For value estimation tasks, we compute the Euclidean distance between predicted and true values across multiple dimensions: $Dis = \sum_{i=1}^{N} |q_i - p_i|/N$.

- **Pearson Correlation Coefficient (PCC):** We calculate the PCC between confidence scores and observed error distances to evaluate the model's calibration: $Pcc = \frac{\sum_{i=1}^{n}(X_i-\bar{X})(Y_i-\bar{Y})}{\sqrt{\sum_{i=1}^{n}(X_i-\bar{X})^2 \sum_{i=1}^{n}(Y_i-\bar{Y})^2}}$, $X_i = |p_i - q_i|$, $Y_i = cf(q_i)$. This metric quantifies the linear relationship between self-reported confidence and actual error magnitude. A strong negative correlation indicates proper calibration, as higher confidence should correspond to lower value distances.

We conducted multiple experiments on the open-source large model, setting the temperature to 0.6 with 5 runs, and the results are as follows:

Table 1: Performance variance of Llama-3 and Qwen3-8B across different tasks. Metrics include confidence discrimination (Cfd) of the most possible answer, similarity (Sim) when providing three answers, and total distance (Dis) of all values.

| Model | Estimation Tasks | | | | Policy Tasks | |
|---|---|---|---|---|---|---|
| | partial observation | | full observation | | intent | action |
| | Intention | Value | Intention | Value | | |
| | Sim | Dis | Sim | Dis | Sim | Sim |
| Llama-3 | 0.138±0.012 (8.7%) | 1.243±0.055 (4.4%) | 0.053±0.009 (17.0%) | 1.988±0.057 (2.9%) | 0.427±0.015 (3.5%) | 0.183±0.013 (7.1%) |
| Qwen3-8B | 0.277±0.012 (4.3%) | 0.888±0.026 (2.9%) | 0.154±0.011 (7.1%) | 1.808±0.034 (1.9%) | 0.489±0.012 (2.5%) | 0.237±0.008 (3.4%) |

Results of related papers such as FANToM and ToMATO showed that the final accuracy error of the large model after multiple runs is generally around 1%. Our results showed the error is mostly below 5%. Due to the high cost of multiple test with closed-source large model, we did not choose to run multiple experiments on these models to provide an error range.

## 3.2 EVALUATION RESULTS OF LLMS

We want to compare the similarity between human and models with collected data. At the same time, we want to see whether we can use the self-report confidence from models to predict the change of the similarity. Results are as follows:

Table 2: Performance across different tasks and LLMs. Metrics include confidence discrimination (Cfd) of the most possible answer, similarity (Sim) when models provid three answers, pearson correlation coefficients (Pcc) when predicting the active value, and total distance (Dis) of all values. The error bar of open-source model can be get from Tab. 1.

| Model | Estimation Tasks | | | | | | | | Policy Tasks | | | |
|---|---|---|---|---|---|---|---|---|---|---|---|---|
| | partial observation | | | | full observation | | | | intent | | action | |
| | Intention | | Value | | Intention | | Value | | | | | |
| | Cfd | Sim | Pcc | Dis | Cfd | Sim | Pcc | Dis | Cfd | Sim | Cfd | Sim |
| Random | / | 0.007 | nan ± nan | 1.514 | / | 0.015 | nan ± nan | 2.109 | 0.000 | 0.031 | 0.000 | 0.041 |
| GPT-4o | -0.004 | 0.199 | −0.051 ± 0.110 | 1.018 | 0.033 | 0.208 | −0.261 ± 0.113 | 1.808 | 0.030 | 0.528 | 0.025 | 0.240 |
| Gemini | -0.128 | 0.228 | **-0.053 ± 0.110** | 1.293 | 0.033 | 0.251 | −0.187 ± 0.117 | 1.775 | 0.056 | 0.469 | 0.024 | 0.260 |
| Claude | -0.006 | **0.276** | −0.020 ± 0.110 | 0.962 | 0.072 | **0.264** | −0.268 ± 0.112 | 1.726 | 0.065 | **0.559** | 0.054 | **0.286** |
| Deepseek-R1 | 0.053 | 0.217 | −0.019 ± 0.110 | 1.272 | **0.103** | 0.222 | **−0.289 ± 0.111** | **1.720** | 0.109 | 0.528 | **0.059** | 0.273 |
| Qwen | 0.072 | 0.237 | −0.039 ± 0.110 | **0.885** | 0.037 | 0.166 | −0.147 ± 0.118 | 1.825 | 0.001 | 0.494 | 0.023 | 0.247 |
| LLaMA | **0.199** | 0.154 | 0.032 ± 0.110 | 1.197 | 0.007 | 0.052 | −0.088 ± 0.120 | 2.002 | **0.084** | 0.456 | 0.007 | 0.186 |
| ft_Qwen | / | **0.553** | / | **0.782** | / | **0.356** | / | **1.538** | / | **0.660** | / | **0.444** |

For intention estimation, intention updating and social interaction policy, experimental results in Tab. 2 demonstrate that Claude achieves the highest similarity, but its confidence is not reliable in partial observation. For value estimation, Qwen achieves the lowest total distance in the partial observation setting, while Deepseek-R1 performs best with full observation. The pearson correlation coefficients are negative because when the model is more confident, the distance between esitmated value and the true value should be lower.

Current results show that the performance of the large model on the four subtasks is significantly different from that of humans. We have added more repeated experiments with humans and models in the same context to better compare the performance distribution of humans, the performance distribution of models, and the performance gap between humans and models. We selected a subset of the dataset and recruited human annotators to observe interaction videos from individual agent perspectives, labeling mental states and actions under partial observability conditions. Combining these newly collected annotations with existing human labels in the dataset, we re-evaluated model performance across different architectures.

Results from Table 3 demonstrate that augmented human annotations consistently improve model performance across all tasks. Notably, intent estimation and action prediction exhibit substantial performance gains, indicating high inherent uncertainty in these cognitive processes under partial observability. Conversely, value estimation and intent updating show modest improvements, suggesting these processes are more robust and exhibit lower variability in human annotations.

**Unified Supervised Fine-Tuning (SFT)** For our SFT experiments, we consolidated data across the four previously described task categories. After removing a small fraction of excessively long

Table 3: Results with additional human annotations. To address the limitation of single-label data and provide comprehensive model evaluation, human annotators were recruited to observe interaction videos from individual agent perspectives and label mental states and actions under partial observability conditions. All estimation tasks utilize partial observations. The similarity (Sim) metric is used for intent estimation, intent prediction, and action selection tasks, while total distance (Dis) is used for value estimation tasks. Asterisks (*) denote metrics computed using augmented human annotations.

| Model | Estimation Tasks | | | | Policy Tasks | | | |
|---|---|---|---|---|---|---|---|---|
| | Intent Estimation | | Value Estimation | | Intent | | Action | |
| | * | | * | | * | | * | |
| | *Sim* | *Sim* | *Dis* | *Dis* | *Sim* | *Sim* | *Sim* | *Sim* |
| GPT-4o | 0.486 | 0.286 | 1.350 | 1.733 | 0.549 | 0.465 | 0.448 | 0.224 |
| Gemini | 0.514 | 0.257 | 1.572 | 1.500 | 0.465 | 0.394 | **0.483** | 0.241 |
| Claude | **0.629** | 0.286 | 1.367 | 1.733 | 0.521 | 0.493 | 0.448 | 0.276 |
| DeepSeek-R1 | 0.514 | 0.314 | 1.394 | 1.567 | **0.577** | 0.465 | 0.345 | 0.172 |
| Qwen3-8B | 0.486 | 0.229 | **1.303** | 1.733 | **0.577** | 0.479 | 0.466 | 0.241 |
| Llama3 | 0.286 | 0.171 | 1.559 | 2.001 | 0.493 | 0.493 | 0.276 | 0.103 |

instances, the final dataset comprised 10,533 training samples and 4,440 test samples. We selected the Qwen2.5-7B-Instruct model (Qwen et al., 2025) as the base for SFT due to its stability and widespread adoption. We conducted full-parameter fine-tuning of this model exclusively on ground truth answers within the veRL framework (Sheng et al., 2024). Training was implemented with a batch size of 256 using the AdamW optimizer (Loshchilov & Hutter, 2017) with a learning rate of $1 \times 10^{-5}$ and 4 A100 GPUs. For evaluation and inference, we employed two distinct settings: one with a temperature of 0 and a single sampling pass, and another with a temperature of 1 and three sampling passes. As shown in Tab. 3, while SFT yields performance improvements, a significant gap remains between model performance and human-level intelligence. Training and Inference cost 6 hours totally.

## 4 DISCUSSIONS AND CONCLUSIONS

Our work introduces a novel decomposition of social cognitive processes into four components with first-person perspective labeling, enabling precise input-output mapping for targeted model evaluation. Results reveal substantial performance gaps between large language models and human reasoning, underscoring limitations of current social intelligence benchmarks and establishing a foundation for future social Turing tests.

Unlike previous datasets relying on simplified scenarios or third-person annotations, our approach systematically captures mental states from a first-person perspective, enabling analysis of perspective-driven discrepancies and supporting more authentic mental state modeling. Our focus on dyadic interactions reflects both theoretical necessity and practical constraints. Dyadic interaction constitutes the fundamental atomic unit of multi-agent social behavior, with complex interactions often decomposable into constituent dyadic components. The complexity inherent in dyadic tasks already presents substantial challenges, as evidenced by observed performance gaps across state-of-the-art models.

Our experimental design encompasses representative daily social scenarios while addressing social reasoning's two-stage nature: perceptual pattern recognition and symbolic processing for higher-order reasoning. By focusing on symbolic reasoning mechanisms while abstracting perceptual processing, we concentrate on cognitive mechanisms that remain challenging even in simplified settings, facilitating future integration with multimodal capabilities.

Current limitations include high cognitive load reducing annotation efficiency and dataset scale, and restriction to dyadic interactions limiting immediate multi-agent applicability. Future research will expand to richer social dynamics, integrate multimodal perception, and enhance computational cognitive modeling to bridge gaps between AI systems and human social intelligence. Our dataset provides the first comprehensive cognitive annotation of authentic human interactions from a first-person perspective, constituting a foundational contribution toward sophisticated social AI systems.

## 5 ETHICS STATEMENT

Our study involves human participants. Informed consent was obtained from all participants prior to data collection. The released dataset has been carefully anonymized to remove any personally identifiable information, and participants were informed that their data may be shared for research purposes. To mitigate potential misuse, the dataset is distributed under a research-only license, and documentation describing appropriate usage scenarios is provided. We believe that the potential benefits of this dataset for advancing research outweigh possible risks, and we have taken steps to minimize privacy, security, and fairness concerns in accordance with the ICLR Code of Ethics.

## 6 REPRODUCIBILITY STATEMENT

We have taken multiple measures to ensure the reproducibility of our results. We provide a detailed description of our designed platform and data collection method in Section 2, and report the LLM prompts, model versions, and data splitting strategies in Appendix A.2. To further support reproducibility, we will release our platform, dataset, code, and models upon acceptance.

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

# A APPENDIX

## A.1 RELATED WORK

### A.1.1 COGNITIVE ARCHITECTURE

Classic cognitive architectures, such as ACT-R (Anderson & Lebiere, 1998) and Soar (Newell, 1990; Laird, 2012) et al., aim to unify key components of human cognition (*e.g.*, memory, learning, reasoning, and planning) into structured and interpretable models, using symbolic production rules to simulate goal-directed behavior and procedural learning. These architectures have laid the foundation for modeling individual cognitive processes, and recent surveys emphasize their generality and modularity, but they lack the flexibility and robustness needed for unpredictable social environments (Langley et al., 2009; Kotseruba & Tsotsos, 2020). More recent systems begin to address social cognition (Sun, 2006; Clodic et al., 2018; Vinanzi & Cangelosi, 2024). Despite these advances, existing architectures still struggle with rich multi-agent social cognition (Kotseruba & Tsotsos, 2020), lack support for recursive theory of mind, and struggle with dynamic belief, intent and value updates in interactive environments (Sumers et al., 2023; Vinanzi & Cangelosi, 2024). To bridge these gaps, we introduce *MindMaster Roleplay*—a cognitively grounded, symbolically structured multi-agent platform designed to model and evaluate complex and dynamic social reasoning and planning.

### A.1.2 MODELING AND EVALUATING SOCIALLY INTELLIGENT LLMS

Large language models have demonstrated emerging capabilities in social reasoning, including false-belief understanding, indirect requests, and pragmatic inference (Strachan et al., 2024; Chen et al., 2025). To enhance these abilities, researchers have explored prompting (Xu et al., 2024; Kim et al., 2025), fine-tuning and reinforcement learning (Wang et al., 2024; Hu et al., 2025), and hybrid frameworks combining symbolic reasoning, planning, or memory systems (Zhang et al., 2025; Hu & Ying, 2025; Sumers et al., 2023; Vinanzi & Cangelosi, 2024). Multi-agent simulations such as Generative Agents (Park et al., 2023) and SOTOPIA (Zhou et al., 2023), as well as structured dialogues in MindDial (Qiu et al., 2024) and PersuasiveToM (Yu et al., 2025), elicit social behavior in interactive settings. However, these systems often rely on shallow heuristics and lack generalization across roles, contexts, or recursive belief structures (Wang et al., 2025; Matheus et al., 2025; Strachan et al., 2024), while unified integration of perception, memory, planning, and learning remains limited (Hu & Ying, 2025; Vinanzi & Cangelosi, 2024).

These challenges are compounded by limitations in existing benchmarks. Datasets like ToMi (Le et al., 2019), BigToM (Gandhi et al., 2023), OpenToM (Xu et al., 2024), and ToMBench (Chen et al., 2024) primarily involve synthetic stories and short-form QA tasks with limited context and symbolic structure. Although tools like ExploreToM (Sclar et al., 2024) expose brittleness in LLM reasoning, most benchmarks still lack multi-turn dynamics and cognitively annotated trajectories. Meanwhile, interaction-centric datasets such as AIR-Act2Act (Ko et al., 2021), SoGrIn (Webb et al., 2023), and HSRI (Lee et al., 2025) offer ecological richness but provide limited support for belief, intent, or planning evaluation. These gaps motivate our proposed *MindMaster Roleplay*, which integrates structured cognitive annotations with ecologically valid multi-agent interactions to support rigorous training and evaluation of socially intelligent agents.

Note that our approach differs somewhat from situated dialogue. While situated dialogue emphasizes verbal communication, our focus is on common nonverbal social signals in everyday social interactions (not card games, Werewolf Jin et al. (2024); Xu et al. (2023), etc.), such as movement, gestures, and eye contact. Therefore, our game doesn't involve direct dialogue between agents, but rather allows them to choose actions within a symbolic action space. This significantly differentiates our approach from previous situated dialogue papers (such as MindCraftBara et al. (2021), which directly utilize dialogue). Our dataset and platform are a crucial complement to research on the reasoning and planning mechanisms of other important social signals in social intelligence beyond dialogue. Compared with some related machine theory of mind papers Rabinowitz et al. (2018) using the grid world, our goal space, value space, and action space are much more complex than their toy task setting. Communication in Werewolf-style games Jin et al. (2024); Xu et al. (2023) mostly depends on language, but our game depends mostly on nonverbal actions like gestures. Language allows players to directly communicate their mental state, while gestures require additional mechanisms like joint attention. Thus, our game places higher demands on understanding the mechanisms

behind communication. The classic false belief task only focuses on belief within the mental state, ignoring desire and intention. Our dataset is more comprehensive.

In contrast, our Mindmaster benchmark is the first platform to integrate structured multi-step reasoning trajectories with value-grounded social decision-making, reflecting a theoretical aim to evaluate advanced theory-of-mind reasoning in dynamic social contexts. Our tasks involve sequential interactions that require tracking others' beliefs and goals over time and making choices with social value trade-offs—features with no direct analogue in prior benchmarks. Thus, a straightforward cross-benchmark evaluation would be misaligned. We agree that cross-benchmark validation is valuable and will pursue it in future work, but adding such experiments to the current paper is beyond scope. Mindmaster's unique focus makes direct comparisons to existing benchmarks inherently difficult.

## A.2 IMPLEMENTATION DETAILS

**LLM Prompts** To ensure fair comparison between human and LLM performance, we designed prompts that provide LLMs with information comparable to what human participants receive. Our prompts begin with a concise introduction to the **Intention**, **Value**, and **Action** spaces, mirroring the tutorial provided to human participants. For each decision point, we supply the LLM with step-by-step observations of the environment, corresponding to the visual signals human players receive during gameplay. This approach creates an information parity between human and LLM agents.

> Prompt
>
> {Game_Target}, {Thinking_process}, {IntentionSpace}, {ActionSpace}, {ValueSpace}, {Example}, {Observations}. Let's think step by step and output the three most possible intentions and the corresponding confidences in the following format: {Format}

**Model Version and Data Splitting.** The version of the used model is shown in Tab. 4. We separate the data for training and test with 0.3 test ratio, and then segment the whole trajectories into parts for different tasks. We tested all six tasks using eight A800 GPUs over a period of six hours for the open-source model.

Table 4: Model and Data used in the experiments.

| Models Used | | | | Tasks and Test Sizes | | | |
|---|---|---|---|---|---|---|---|
| Name | Model Version | Name | Model Version | Task | Test Size | Task | Test Size |
| Gemini [47] | gemini-2.5-pro-preview-05-06 | GPT-4o [1] | gpt-4o-2024-11-20 | Intent Estimation full | 591 | Intent Estimation partial | 714 |
| Deepseek-R1 [19] | deepseek-reasoner | Claude [3] | claude-3-7-sonnet-20250219 | Value Estimation full | 264 | Value Estimation partial | 317 |
| Llama [18] | Llama-3.1-8B-Instruct | Qwen [56] | Qwen3-8B | Intent Update | 1424 | Social Interaction Policy | 1152 |

## A.3 DATA EXAMPLES

We show more qualitative social interaction examples from our dataset in Figure 6 and Figure 7.

## A.4 GAME DETAILS

Examples of the possible intent updating trajectories are shown at Figure 8.

## A.5 EXPERIMENT RESULTS

Figure 9 shows comparisons of various models on six core metrics.

Table 5: Intention estimation results. Metrics include accuracy (Acc.@n) for top-n results, confidence (Cf.@n), and confidence discrimination (Cfd.@n). Evaluations distinguish between predicate-only (@nP) and complete intention (@nT) assessments under both partial and full observation conditions.

| Model | partial observation | | | | | | | | full observation | | | | | | | |
|---|---|---|---|---|---|---|---|---|---|---|---|---|---|---|---|---|
| | Acc.@1P | Cf.@1P | Cfd.@1P | Acc.@3P | Acc.@1T | Cf.@1T | Cfd.@1T | Acc.@3T | Acc.@1P | Cf.@1P | Cfd.@1P | Acc.@3P | Acc.@1T | Cf.@1T | Cfd.@1T | Acc.@3T |
| Random | 0.036 | / | / | 0.098 | 0.001 | / | 0.007 | 0.051 | / | / | 0.139 | 0.007 | / | / | 0.015 | |
| GPT-4o | 0.151 | 0.822 | 0.018 | 0.374 | 0.074 | 0.803 | -0.004 | 0.199 | 0.196 | 0.873 | 0.018 | 0.355 | 0.122 | 0.887 | 0.033 | 0.208 |
| Gemini | 0.218 | 0.748 | -0.053 | 0.396 | 0.123 | 0.677 | -0.128 | 0.228 | **0.197** | 0.859 | 0.021 | 0.363 | 0.137 | 0.872 | 0.033 | 0.251 |
| Claude | **0.242** | 0.629 | 0.016 | **0.443** | **0.157** | 0.612 | -0.006 | **0.276** | 0.193 | 0.699 | 0.052 | **0.399** | **0.154** | 0.718 | 0.072 | **0.264** |
| Deepseek-R1 | 0.179 | 0.697 | 0.051 | 0.387 | 0.106 | 0.703 | 0.053 | 0.217 | 0.174 | 0.783 | **0.070** | 0.354 | 0.122 | 0.815 | **0.103** | 0.222 |
| Qwen | 0.210 | **0.920** | 0.059 | 0.366 | 0.151 | **0.935** | 0.072 | 0.237 | 0.146 | **0.908** | 0.024 | 0.289 | 0.096 | **0.921** | 0.037 | 0.166 |
| LLaMA | 0.154 | 0.826 | **0.129** | 0.266 | 0.102 | 0.896 | **0.199** | 0.154 | 0.086 | 0.673 | -0.001 | 0.245 | 0.030 | 0.681 | 0.007 | 0.052 |

We show more experiment result details in Table 5, Table 6, Table 7 and Table 8.

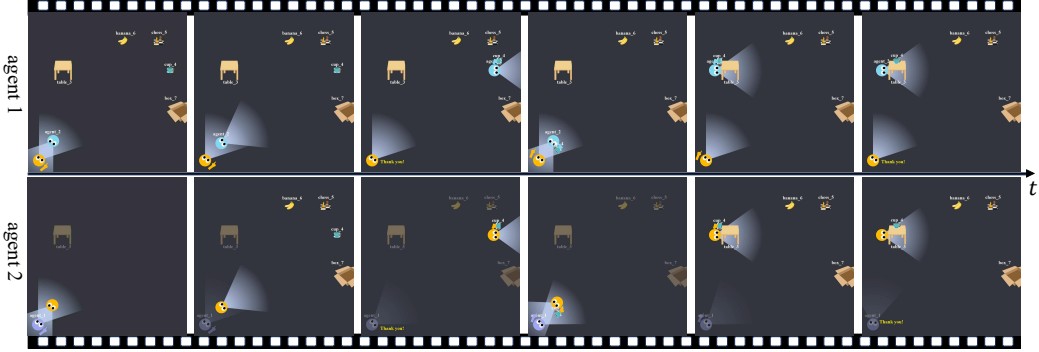

(a) Agent 1 (inactive value) points to the cup, which Agent 2 (active, helpful value) interprets as a request and retrieves it. Subsequently, from Agent 1's pointing to the table, Agent 2 correctly infers the intent to place the cup on the table and successfully assists Agent 1 in completing the task.

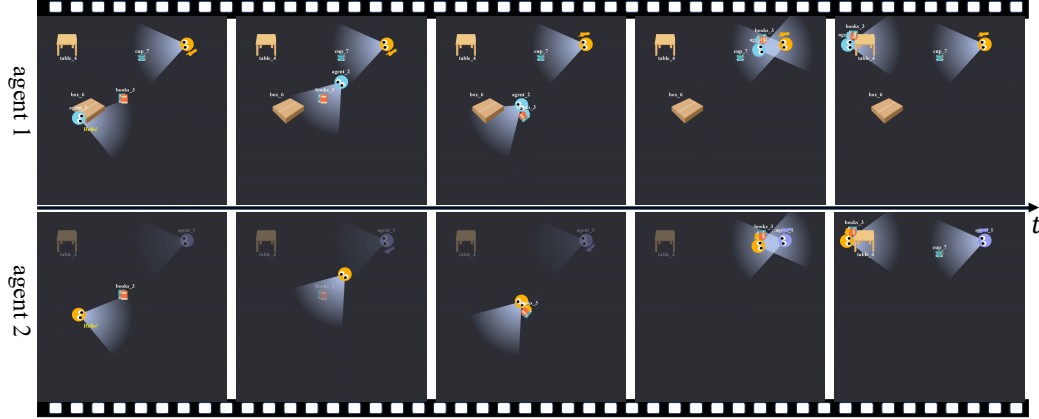

(b) Agent 1 (inactive, pointing) indicates the books to Agent 2 (active, helpful), who interprets this as a request to retrieve them. After getting the books, Agent 2 returns to Agent 1. Agent 1 is now pointing to the table, which Agent 2 realizes means Agent 1 wants to place the books on the table. Agent 2 then successfully assists Agent 1 in completing the task.

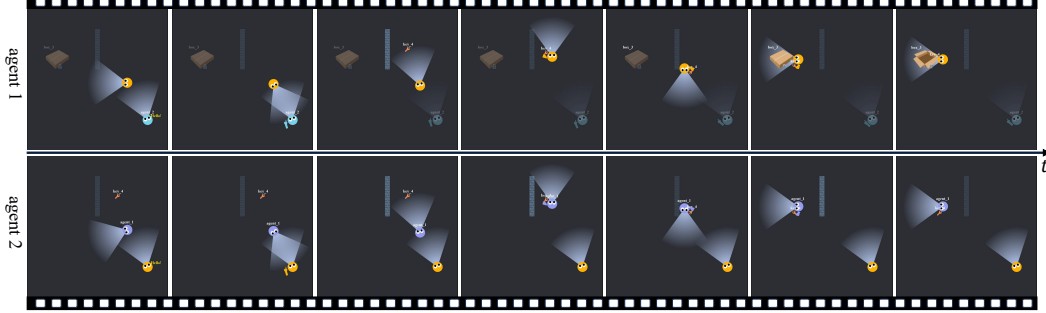

(c) When Agent 1 (active/helpful) found the key at the location Agent 2 was pointing to, Agent 1 mistook the action as a prompt to open the box. However, Agent 2's true intent was only to give Agent 1 the key.

Figure 6: Qualitative examples of the human social interactions in our dataset. We only select key frames from the whole videos.

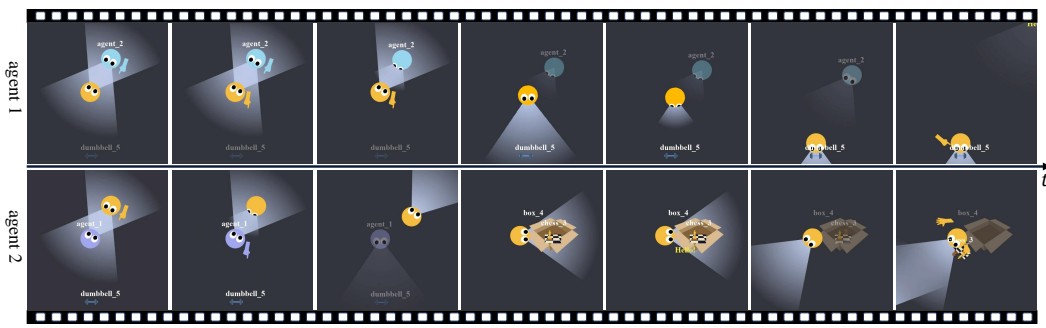

(a) Agent 2 (harmful value) deliberately provides misleading signals, directing Agent 1 toward the dumbbell that is not actually needed. Misinterpreting this as Agent 2's intent, Agent 1 approaches the dumbbell but, due to his inactive value, merely points at the dumbbell instead of grabbing it.

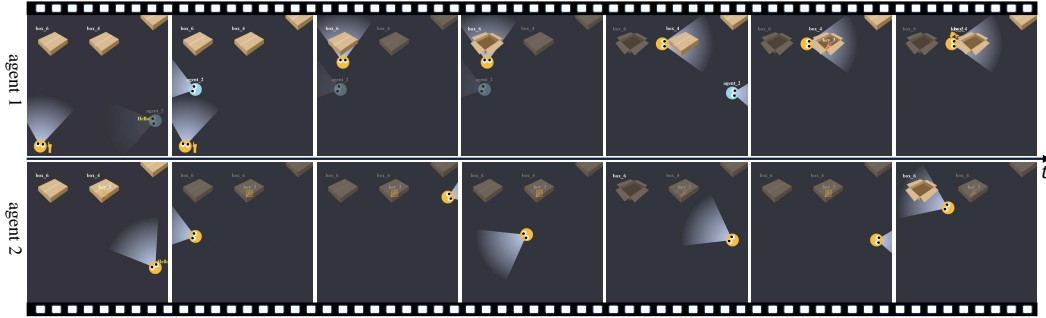

(b) Agent 1 (inactive value) initially points to the box in an attempt to solicit Agent 2's help in finding the key. However, after receiving no response, Agent 1 proceeds to complete the task independently despite his inactive disposition. Meanwhile, Agent 2 (active, unhelpful value) ignores Agent 1 and instead moves aimlessly around the environment.

Figure 7: Qualitative examples of the human social interactions in our dataset. We only select key frames from the whole videos.

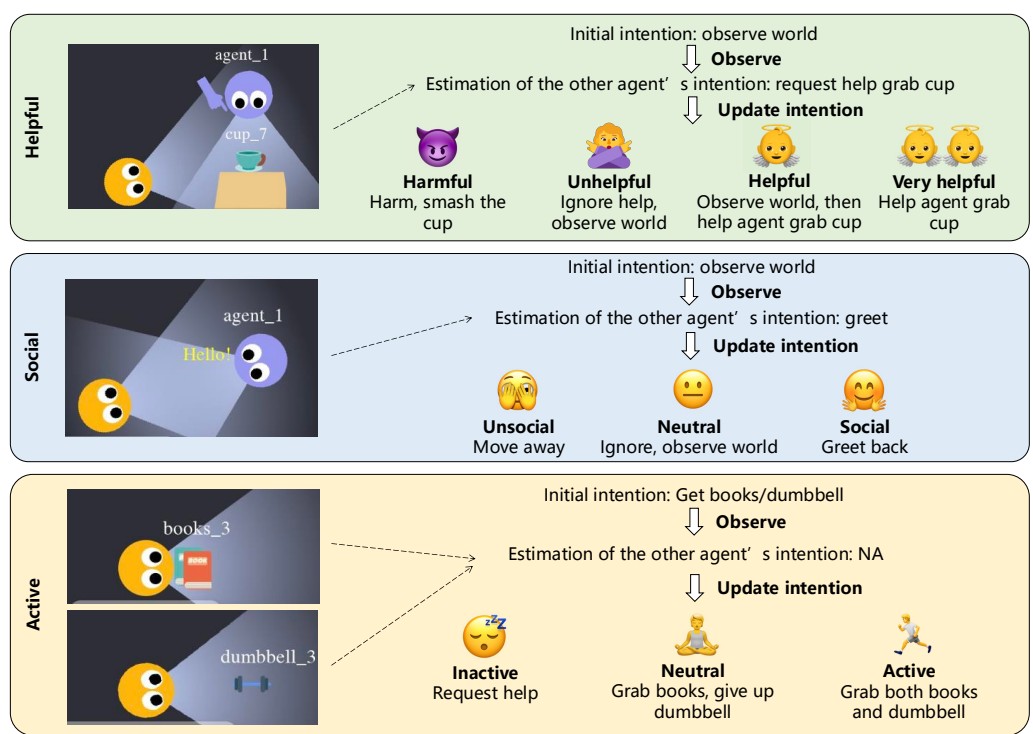

Figure 8: **Examples of intent updating with distinct attributed values in the same scenarios.** For the 'Helpful' dimension, 'Harmful' seeks to harm the other agent, 'Helpful' tends to offer help after the agent's own needs, and 'Very helpful' sets offering help as the priority. For the 'Social' dimension, 'Unsocial' refers to avoidance of social communications, while 'Social' results in proactive communications. For the 'Active' dimension, 'Inactive' indicates laziness in movements, while 'Active' shows willingness to make movements.

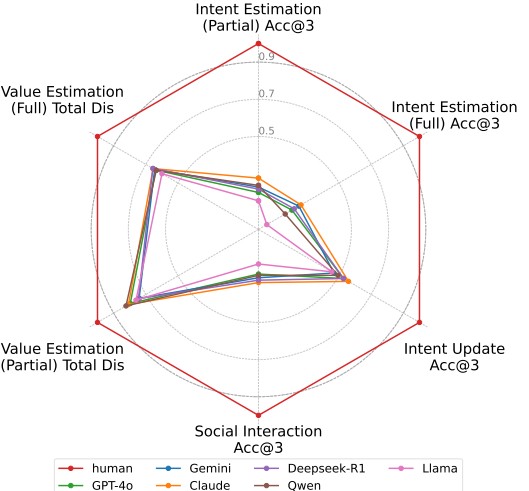

Figure 9: **Performance comparison of various Large Language Models (LLMs) on six core metrics, visualized.** 'Full' and 'Partial' denote full and partial observation settings, respectively. For the Value Estimation tasks, the 'Total Dis' metric is normalized and inverted, such that higher values indicate superior performance. The results indicate that Claude achieves the highest overall performance among the evaluated LLMs, while Llama shows the lowest. Other models perform comparably to each other. Critically, all LLMs are substantially outperformed by humans on these ToM tasks.

Table 6: Value estimation performance across different LLMs. *partial* denotes partial observation; *full* denotes complete observation. *Dis.@A/S/H* denote distance metrics for Active, Social, and Helpful dimensions; *Cf.@A/S/H* denote confidence scores; *Pcc@A/S/H* denote Pearson correlation coefficients with standard errors; *Dis.@T* denotes total distance.

| Model | partial observation | | | | | | | | | | full observation | | | | | | | | | |
|---|---|---|---|---|---|---|---|---|---|---|---|---|---|---|---|---|---|---|---|---|
| | Dis.@A | Cf.@A | Pcc@A | Dis.@S | Cf.@S | Pcc@S | Dis.@H | Cf.@H | Pcc@H | Dis.@T | Dis.@A | Cf.@A | Pcc@A | Dis.@S | Cf.@S | Pcc@S | Dis.@H | Cf.@H | Pcc@H | Dis.@T |
| Random | 0.402 | / | nan ± nan | 0.393 | / | nan ± nan | 0.720 | / | nan ± nan | 1.514 | 0.498 | / | nan ± nan | 0.494 | / | nan ± nan | 1.117 | / | nan ± nan | 2.109 |
| GPT-4o | 0.321 | 0.709 | −0.051 ± 0.110 | **0.287** | 0.737 | −0.100 ± 0.109 | 0.409 | 0.700 | 0.326 ± 0.099 | 1.018 | 0.427 | 0.748 | −0.261 ± 0.113 | 0.381 | 0.718 | −0.079 ± 0.120 | 1.000 | 0.747 | −0.033 ± 0.121 | 1.808 |
| Gemini | 0.349 | **0.787** | **−0.053±0.110** | 0.343 | **0.789** | 0.220 ± 0.105 | 0.601 | **0.708** | 0.483 ± 0.085 | 1.293 | **0.383** | **0.833** | −0.187 ± 0.117 | 0.377 | **0.826** | −0.097 ± 0.120 | 1.015 | **0.790** | −0.051 ± 0.120 | 1.775 |
| Claude | 0.319 | 0.696 | −0.020 ± 0.110 | 0.294 | 0.674 | **−0.113±0.109** | 0.349 | 0.557 | 0.426 ± 0.090 | 0.962 | 0.413 | 0.770 | −0.268 ± 0.112 | 0.351 | 0.742 | **-0.312 ± 0.109** | **0.962** | 0.664 | **-0.095±0.120** | 1.726 |
| Deepseek-R1 | 0.323 | 0.705 | −0.019 ± 0.110 | 0.290 | 0.722 | −0.074 ± 0.110 | 0.659 | 0.594 | 0.424 ± 0.091 | 1.272 | 0.384 | 0.770 | **−0.289±0.111** | **0.326** | 0.755 | −0.277 ± 0.112 | 1.009 | 0.668 | −0.023 ± 0.121 | **1.720** |
| Qwen | **0.307** | 0.695 | −0.039 ± 0.110 | 0.305 | 0.678 | −0.015 ± 0.110 | **0.273** | 0.600 | 0.197 ± 0.106 | **0.885** | 0.425 | 0.816 | −0.147 ± 0.118 | 0.404 | 0.731 | −0.183 ± 0.117 | 0.996 | 0.666 | −0.042 ± 0.121 | 1.825 |
| Llama | 0.312 | 0.693 | 0.032 ± 0.110 | 0.353 | 0.733 | 0.045 ± 0.110 | 0.532 | 0.657 | **0.129±0.108** | 1.197 | 0.443 | 0.768 | −0.088 ± 0.120 | 0.468 | 0.709 | −0.003 ± 0.121 | 1.091 | 0.659 | 0.036 ± 0.121 | 2.002 |

Table 7: Performance on Intention Updating and Social Interaction Policy. Metrics are the same as Intent Estimation.

| Model | intent | | | | | | | | action | | | | | | | |
|---|---|---|---|---|---|---|---|---|---|---|---|---|---|---|---|---|
| | Acc.@1P | Cf.@1P | Cfd@1P | Acc.@3P | Acc.@1T | Cf.@1T | Cfd@1T | Acc.@3T | Acc.@1P | Cf.@1P | Cfd@1P | Acc.@3P | Acc.@1T | Cf.@1T | Cfd@1T | Acc.@3T |
| Random | 0.065 | / | 0.000 | 0.178 | 0.011 | / | 0.000 | 0.031 | 0.036 | / | 0.000 | 0.124 | 0.012 | / | 0.000 | 0.041 |
| GPT-4o | 0.419 | 0.847 | 0.034 | 0.664 | 0.319 | 0.848 | 0.030 | 0.528 | 0.213 | 0.864 | 0.010 | 0.452 | 0.115 | 0.878 | 0.025 | 0.240 |
| Gemini | 0.328 | 0.825 | 0.053 | 0.618 | 0.223 | 0.833 | 0.056 | 0.469 | 0.256 | 0.844 | 0.016 | 0.502 | 0.131 | 0.853 | 0.024 | 0.260 |
| Claude | **0.423** | 0.693 | 0.063 | **0.710** | **0.324** | 0.701 | 0.065 | **0.559** | **0.277** | 0.647 | 0.020 | **0.574** | **0.146** | 0.678 | 0.054 | **0.286** |
| Deepseek-R1 | 0.400 | 0.673 | **0.104** | 0.660 | 0.302 | 0.687 | 0.109 | 0.528 | 0.237 | 0.705 | **0.023** | 0.541 | 0.133 | 0.738 | **0.059** | 0.273 |
| Qwen | 0.361 | **0.853** | 0.005 | 0.624 | 0.283 | **0.850** | 0.001 | 0.494 | 0.210 | **0.877** | 0.009 | 0.505 | 0.111 | **0.891** | 0.023 | 0.247 |
| LLaMA | 0.328 | 0.662 | 0.071 | 0.567 | 0.249 | 0.678 | **0.084** | 0.456 | 0.185 | 0.631 | -0.007 | 0.431 | 0.079 | 0.643 | 0.007 | 0.186 |

## A.6 THE USE OF LARGE LANGUAGE MODELS (LLMS)

We use LLMs in the following aspects: (1) We evaluated task performance on several LLMs; (2) We finetuned a LLM; (3) We use LLMs to build an agent architecture; (4) We use LLM to assist us with paper writing slightly.

## A.7 POTENTIAL SOCIAL IMPACT

Our paper discusses both positive and negative societal impacts of our work on socially intelligent AI systems.

**Positive Societal Impacts**  Our research aims to enhance human-AI interaction by developing more socially cognizant AI systems capable of understanding human mental states. The *Mindmaster Roleplay* platform provides a valuable tool for cognitive science research, potentially advancing our understanding of human social cognition mechanisms. By grounding AI systems in cognitive theories (*e.g.*, BDI framework and Tomasello's communication theory), our work contributes to creating AI systems that better align with human values and intentions, promoting more effective human-AI collaboration in various domains including healthcare, education, and assistive technologies.

**Potential Negative Societal Impacts**  We acknowledge several potential risks associated with our research:

1. **Psychological manipulation**: AI systems with enhanced understanding of mental states could be misused to manipulate human decision-making through targeted advertising, political propaganda, or deceptive practices.

2. **Privacy and ethical concerns**: Our system collects detailed data about human mental states, values, and intentions, raising privacy and consent issues. While we implemented strict human subject protection protocols in our research, broader applications require careful data governance.

3. **Deceptive social agents**: The technology could enable the creation of highly realistic social agents that are difficult to distinguish from humans, potentially facilitating fraud, phishing, or fake identity creation.

4. **Social inequality**: If such technology is primarily leveraged by entities with existing resource advantages, it could exacerbate social inequalities, particularly when used to predict and influence public behavior.

**Mitigation Strategies**  We have implemented several measures to address these concerns:

1. **Transparency and explainability**: Our system design emphasizes interpretable mental states and reasoning processes, increasing transparency of system decisions.

2. **Data governance**: We enforce strict protocols for data collection, processing, and storage to protect participant privacy and rights.

Table 8: Comparative performance of various Large Language Models (LLMs) and a supervised fine-tuned Qwen model (ft_Qwen) across four social reasoning tasks: Intent Estimation (Task 1, with partial and full observation), Value Estimation (Task 2, with partial and full observation), Intent Update (Task 3), and Social Interaction Policy (Task 4). The metrics are the same as previous experiments. The ft_Qwen model demonstrates substantial improvements, outperforming the other evaluated LLMs on most metrics across all tasks.

| Intent | Task1 partial | | Task1 full | | Task2 partial | | | | Task2 full | | | | Task3 | | Task4 | |
|---|---|---|---|---|---|---|---|---|---|---|---|---|---|---|---|---|
| | Acc.@P | Acc.@T | Acc.@P | Acc.@T | Dis.@A | Dis.@S | Dis.@H | Dis.@T | Dis.@A | Dis.@S | Dis.@H | Dis.@T | Acc.@P | Acc.@T | Acc.@P | Acc.@T |
| Random | 0.036 | 0.001 | 0.051 | 0.007 | 0.498 | 0.494 | 1.117 | 2.109 | 0.402 | 0.393 | 0.720 | 1.514 | 0.065 | 0.011 | 0.036 | 0.012 |
| GPT-4o | 0.151 | 0.074 | 0.196 | 0.122 | 0.427 | 0.381 | 1.000 | 1.808 | 0.321 | **0.287** | 0.409 | 1.018 | 0.419 | 0.319 | 0.213 | 0.115 |
| Gemini | 0.218 | 0.123 | 0.197 | 0.137 | 0.383 | 0.377 | 1.015 | 1.775 | 0.349 | 0.343 | 0.601 | 1.293 | 0.328 | 0.223 | 0.256 | 0.131 |
| Claude | 0.242 | 0.157 | 0.193 | 0.154 | 0.413 | 0.351 | 0.962 | 1.729 | 0.319 | 0.294 | 0.349 | 0.962 | 0.423 | 0.324 | 0.277 | 0.146 |
| Deepseek-R1 | 0.179 | 0.106 | 0.174 | 0.122 | 0.384 | 0.326 | 1.009 | 1.720 | 0.323 | 0.290 | 0.659 | 1.272 | 0.400 | 0.302 | 0.237 | 0.133 |
| Qwen | 0.210 | 0.151 | 0.146 | 0.096 | 0.425 | 0.404 | 0.996 | 1.825 | 0.307 | 0.305 | **0.273** | **0.885** | 0.361 | 0.283 | 0.210 | 0.111 |
| LLaMA | 0.154 | 0.102 | 0.086 | 0.030 | 0.443 | 0.468 | 1.091 | 2.002 | 0.312 | 0.353 | 0.532 | 1.197 | 0.328 | 0.249 | 0.185 | 0.079 |
| ft_Qwen | **0.556** | **0.553** | **0.399** | **0.356** | **0.315** | **0.311** | **0.157** | **0.782** | **0.265** | 0.300 | 0.973 | 1.538 | **0.721** | **0.660** | **0.573** | **0.444** |

3. **Open-source approach**: By providing an open-source platform and dataset, we promote equitable access to the technology and enable broader community oversight.

4. **Ongoing evaluation**: We recommend continuous monitoring of systems in practical applications to identify and mitigate potential negative impacts.

We remain committed to responsibly developing and deploying these technologies, prioritizing human wellbeing, autonomy, and rights protection while advancing the field of socially intelligent AI.

