# OpenReview forum: "Mindmaster Roleplay: A Social Reasoning and Planning Benchmark"
_ICLR.cc/2026/Conference — ICLR 2026 Conference Desk Rejected Submission_

### Official Review · Reviewer_nTGn · 2025-10-28

**Soundness:** 2
**Presentation:** 3
**Contribution:** 2
**Rating:** 4
**Confidence:** 4

**Summary:**

This paper introduces Mindmaster Roleplay, a dyadic role-playing platform for studying social reasoning and planning. It integrates belief–intention–value modeling within interactive simulations and collects first-person mental-state annotations from 232 human participants. The authors benchmark several LLMs on four subtasks and release a cognitively grounded dataset for evaluating social intelligence.

**Strengths:**

1. Tackles the important but underexplored challenge of social reasoning in AI, focusing on dynamic mental-state modeling (belief, intent, value).

2. Presents an interpretable, physically simulated dyadic environment—an improvement over static text-based ToM benchmarks.

3. Collects first-person cognitive annotations from 232 participants with strong quality control, yielding interpretable, scientifically grounded data.

4. Decomposes social cognition into four subtasks and evaluates multiple major LLMs under partial and full observability.

**Weaknesses:**

1. The paper mainly repackages existing cognitive concepts (BDI, Bayesian ToM, Tomasello’s theory) into a single framework rather than introducing new theoretical or algorithmic contributions. The innovation is incremental rather than conceptual.

2. The benchmark focuses solely on dyadic (two-agent) interactions, which limits its ecological validity. Real-world social reasoning often involves multi-agent dynamics such as coalition formation, shared attention, and group-level decision making. While dyadic settings are easier to control experimentally, this design choice constrains the benchmark’s ability to capture the full spectrum of social cognition.

3. The evaluation relies on simple agreement-based metrics (Sim, Dis, Cfd, PCC) that measure output similarity rather than reasoning quality. These metrics do not capture deeper Theory-of-Mind processes such as recursive belief modeling or second-order reasoning.

4. Experiments are largely descriptive. The paper reports numeric metrics (Sim, Dis, Cfd) but does not analyze why models fail, which reasoning steps are difficult, or what errors occur under partial observability. The results mostly show that “models perform worse than humans,” which is expected. There is little new insight into how reasoning failures manifest or differ across architectures.

5. Beyond benchmarking, the paper claims to “reveal intriguing phenomena in human reasoning,” but such phenomena are not explicitly reported or analyzed. There is limited discussion on what the dataset tells us about human cognition or how these insights could guide model design.

**Questions:**

How do you ensure inter-annotator agreement and minimize bias in first-person mental-state labeling?

Could you provide qualitative examples of model reasoning errors or value conflicts to support your conclusions?

Do the designed social scenarios (e.g., helping, deception, cooperation) generalize well beyond the controlled game environment to more naturalistic or multimodal settings?

---

### Official Review · Reviewer_6UCe · 2025-10-30

**Soundness:** 3
**Presentation:** 2
**Contribution:** 2
**Rating:** 4
**Confidence:** 2

**Summary:**

This paper presents a novel and ambitious attempt to model social reasoning in AI systems through a controlled dyadic role-play platform. The proposed benchmark, Mindmaster Roleplay, builds a structured environment with belief–intention–value representations and collects first-person annotations from 232 human participants.
The authors decompose social cognition into four subtasks (intention/value estimation, updating, and social policy generation) and evaluate a range of LLMs and fine-tuned variants on these components. Overall, the work provides a technically interesting benchmark with real human data and a cognitively interpretable design.

To ACs:
I am not a social scientist, so I may not be the best person to evaluate the sociological or psychological contribution of this work. I write the review from a computer science and AI benchmarking perspective.

**Strengths:**

1. Novel benchmark design. The proposed role-play environment is original and differs from standard text-based ToM or dialogue datasets. It explicitly models belief, intention, and value and captures structured trajectories, which is methodologically interesting.

2. Real human data. The dataset is collected from 232 offline participants with annotated first-person mental states, which is rare among social reasoning benchmarks and adds credibility to the data quality.

**Weaknesses:**

1. Limited environment complexity. The current design has very limited states and actions. Many transitions could be represented by simple hard-coded rules. It is therefore unclear how much genuine reasoning is required, and whether model performance differences truly reflect “social intelligence.”

2. Task transfer and utility. The paper does not analyze whether learning from this dataset can improve LLM performance on other reasoning or interaction tasks. The benefit of fine-tuning on this benchmark (beyond reproducing in-domain metrics) remains unclear.

3. Representation and visualization issues. Some figures, especially Figure 2, though visually appealing, are not informative. The icons and layout do not clearly convey the relationship between actions, intentions, and scenarios, making it hard to understand the full game content.

4. Lack of Evaluation justification. While multiple LLMs are compared, the evaluation mainly reports accuracy-like metrics. It is not clear why these scores correspond to real social reasoning ability rather than pattern recognition over small discrete state spaces.

**Questions:**

1. Given that the number of states and actions is small and the rules are deterministic, how can we be sure that models are reasoning rather than memorizing or mapping from observation to action through hard rules?

2. What concrete benefit does supervised fine-tuning on such constrained role-play data bring to LLMs in broader reasoning or dialogue settings? Have you observed any transfer to open-ended or natural social reasoning tasks?

---

### Official Review · Reviewer_3pJZ · 2025-11-01

**Soundness:** 2
**Presentation:** 2
**Contribution:** 2
**Rating:** 2
**Confidence:** 3

**Summary:**

The paper presents Mindmaster Roleplay, a platform designed to study social reasoning and planning in AI agents through role-playing games. The platform aims to simulate complex social interactions by capturing the interplay between beliefs, intentions, values, and actions in dyadic interactions. It provides a dataset of annotated human social interactions, focusing on first-person perspectives and cognitive traces, which can be used to train and evaluate AI models for social intelligence. The authors evaluate the performance of state-of-the-art large language models against human performance, highlighting the significant gap in social reasoning tasks.

**Strengths:**

Clarity: The paper provides sufficient details about the platform's design and experimental setup, though there are areas where more explanation would enhance clarity, especially regarding the connection between the platform's design and real-world social reasoning.

Significance: The work contributes to AI's understanding of social intelligence by providing a tool for evaluating models on complex, multi-turn social interactions. However, the significance of the work is limited by the lack of innovative technical advancements or new methodologies.

**Weaknesses:**

1. Unclear Relationship Between Game Design and Social Reasoning
The primary issue with this paper is the lack of clarity regarding how the game design relates to real-world social reasoning and planning. Although the paper describes the game’s scenes and tasks in detail, it is not fully explained how these designs genuinely capture human-like social reasoning in real-world contexts. While the game’s rules and mechanics are not overly simplistic, the connection between these behaviors and authentic social reasoning is unclear. The paper does not provide sufficient justification for why these specific tasks are representative of true social reasoning abilities.


2. Lack of Technical Innovation
Overall, the paper feels more like a platform simulation and data collection effort rather than a contribution of novel technical approaches. The focus is primarily on the game design and the dataset collection, with little to no advancement in terms of technical methods or innovative models.  The work lacks a meaningful technical contribution to AI’s social reasoning capabilities.


3. Unclear Practical Value of the Data
I struggle to understand the practical value of the data collected. From the supplementary materials provided, it’s not clear how this rough simulation could have a significant impact on social reasoning research. The data may not offer much beyond basic social interaction modeling, and it’s difficult to see why it would be valuable to researchers or practitioners in this space.

**Questions:**

1. Why impose predefined character roles on participants?

The game’s reliance on predefined roles feels artificial and detached from real-world scenarios. Why is it necessary to assign participants specific character intentions and values from the start? This setup does not reflect how people naturally reason or plan in social contexts, where roles and behaviors emerge more organically from the interaction itself. How does this forced framework align with the goal of studying authentic social reasoning and planning?

2. Does inferring intentions under these predefined roles simplify the task?

Given the constraints imposed by the fixed character roles, it seems that inferring another player’s intentions becomes a simplified behavior mapping task. I struggle to see the deeper design or thought behind the actions and intentions within the game. Are the action choices and reasoning steps genuinely reflective of the complexity in social reasoning, or are they rather simplistic decisions based on the pre-given roles?

3. What is the real value of the data beyond game performance?

Beyond improving the performance of your game and platform, does the dataset offer any significant value? For instance, models trained on this data may simply achieve minor fitting specific to the tasks in your game, without showing meaningful improvement in general social reasoning tasks. Would models trained on this dataset exhibit any meaningful transferability to real-world scenarios, or would their capabilities be limited to the specifics of your platform without contributing to broader advances in social reasoning?

**Details Of Ethics Concerns:**

The supplementary material, 'Turtorial_quiz.pdf', appears to retain the authors' email addresses, which could potentially violate the double-blind review process. This raises concerns regarding the anonymity of the review process

---

> ### Comment · Area_Chair_asnv · 2025-11-26
>
> Could you elaborate on which specific pages to reveal the identity? Thanks!

---

> > ### Author Response · Authors · 2025-11-27
> >
> > The email address mm-roleplay@outlook.com appears in the tutorial_quiz section of the supplementary material. However, please note that this is a generic public account, and no private information has been compromised.

---

### Official Review · Reviewer_V2FM · 2025-11-02

**Soundness:** 3
**Presentation:** 3
**Contribution:** 3
**Rating:** 6
**Confidence:** 3

**Summary:**

This paper presents Mindmaster Roleplay, a new benchmark and platform for studying social reasoning and planning through dyadic role-play games. The environment models agents with explicit belief, intention, and value states, and decomposes social cognition into steps of observing, inferring others’ mental states, updating intentions, and choosing actions. The authors collect a human dataset from 232 participants who self-annotate their mental states and reasoning, and evaluate multiple LLMs (GPT-4o, Claude, Gemini, DeepSeek-R1, Qwen3-8B, Llama-3) plus a fine-tuned Qwen2.5-7B model. Humans significantly outperform all models, especially under partial observability, showing the difficulty of human-like social reasoning.

**Strengths:**

1 Proposes a cognitively grounded platform that explicitly represents belief, intention, and value, going beyond simple ToM question-answering.

2 First-person mental state annotations make the dataset unique and interpretable.

3 Covers non-verbal social reasoning (pointing, gestures, cooperation, deception), a relatively unexplored area.

4 Provides structured subtasks (intention/value estimation, intention updating, policy selection) and multi-model evaluation.

5 Conducts real human experiments with tutorials and quality control, ensuring data reliability.

**Weaknesses:**

1 Questionable ground truth for “mental states.”
Participants explicitly report their own and others’ intentions, but such self-reports are often rationalized or biased by the role assignment (“you are the helpful player”). This may not reflect natural cognition.

2 Simplified value and communication space.
The three value axes (active, social, helpful) and gesture-based interactions are elegant but unrealistic. Real social reasoning involves richer motives, verbal communication, and pragmatic subtleties.

3 Evaluation remains shallow.
The analysis only reports task-level accuracy and calibration, without deeper error diagnosis or temporal dynamics. It is unclear why LLMs fail or which reasoning stages cause errors.

4 Fine-tuning details unclear.
The SFT experiment lacks enough specification (prompt format, task mixing, hyperparameter rationale) for full reproducibility.

**Questions:**

The same as the weaknesses

---

### Note · Program_Chairs · 2026-01-17
**Submission Desk Rejected by Program Chairs**

The following references in this submission do not refer to real documents and/or have major errors in bibliographic information:

 Aurélien Clodic, Aurélien Clodic, and Rachid Alami. An interaction-oriented cognitive and affective architecture for human-robot interaction. In IEEE International Symposium on Robot and Human Interactive Communication (RO-MAN), 2018. 1, 15